# The global burden of yellow fever

Katy AM Gaythorpe[1]*, Arran Hamlet[1], Kévin Jean[2], Daniel Garkauskas Ramos[3], Laurence Cibrelus[4], Tini Garske[1], Neil Ferguson[1]

[1]WHO Collaborating Centre for Infectious Disease Modelling, MRC Centre for Global Infectious Disease Analysis, Abdul Latif Jameel Institute for Disease and Emergency Analytics (J-IDEA), Imperial College London, London, United Kingdom; [2]Maître de conférences, Laboratoire MESuRS - Cnam Paris, Paris, France; [3]Secretariat for Health Surveillance, Brazilian Ministry of Health, Brasilia, Brazil; [4]World Health Organisation, Geneva, Switzerland

**Abstract** Yellow fever (YF) is a viral, vector-borne, haemorrhagic fever endemic in tropical regions of Africa and South America. The vaccine for YF is considered safe and effective, but intervention strategies need to be optimised; one of the tools for this is mathematical modelling. We refine and expand an existing modelling framework for Africa to account for transmission in South America. We fit to YF occurrence and serology data. We then estimate the subnational forces of infection for the entire endemic region. Finally, using demographic and vaccination data, we examine the impact of vaccination activities. We estimate that there were 109,000 (95% credible interval [CrI] [67,000–173,000]) severe infections and 51,000 (95% CrI [31,000–82,000]) deaths due to YF in Africa and South America in 2018. We find that mass vaccination activities in Africa reduced deaths by 47% (95% CrI [10%–77%]). This methodology allows us to evaluate the effectiveness of vaccination and illustrates the need for continued vigilance and surveillance of YF.

## Introduction

Yellow fever is a flavivirus endemic in tropical regions of Africa and South America. In Africa, it is the third most commonly reported type of disease outbreak. In the Americas, yellow fever produces extensive epizootics in non-human primates (NHPs) and outbreaks of human cases (*Mboussou et al., 2019*). It is vaccine preventable, with a safe and effective vaccine available since the 1930s that has been introduced into the Expanded Programme on Immunisation (EPI) of many countries (*Region V, 2003*). Yellow fever is transmitted by numerous vectors including Aedes spp. and Haemogogus spp. in Africa and the Americas, respectively. A component of the sylvatic reservoir system is in NHPs, and as a result of this, yellow fever cannot be eradicated. The clinical course of yellow fever infection leads to a variety of non-specific symptoms with severe infections potentially exhibiting fever, nausea, vomiting, jaundice, and haemorrhaging, which can result in death (*Monath and Vasconcelos, 2015*).

The transmission dynamics of yellow fever have numerous components. There are two main 'cycles' of transmission: urban and sylvatic. The sylvatic cycle is said to be the driver of most reported transmission with infection occurring mainly between NHPs, mediated by tree-hole breeding mosquitos. These vectors are diurnal and feed mostly on NHP; however, people can be infected if they encroach on this cycle through occupational or recreational activities (*Monath and Vasconcelos, 2015*). In South America, this accounts for the majority of cases and can potentially lead to large outbreaks; a recent yellow fever season saw over 1000 cases in Brazil alone (*Couto-Lima et al., 2017*). The urban cycle of yellow fever is less common, but the outbreaks have the potential to be devastating. Whilst urban outbreaks have largely been eradicated in South America (*Câmara et al., 2011*), they still occur in Africa with a recent urban outbreak, in Angola and the Democratic Republic

*For correspondence:
k.gaythorpe@imperial.ac.uk

Competing interests: The authors declare that no competing interests exist.

of the Congo, causing 962 reported cases. Although this is thought to be only a fraction of the actual transmission (*Organization, WH, 2017*). Large outbreaks as a result of urban transmission are due to the combination of densely populated urban areas, and large populations of *Aedes aegypti*, which bites humans preferentially and breeds rapidly in urban environments (*Harrington et al., 2014*). The World Health Organisation (WHO) developed the Eliminate Yellow fever Epidemics (EYE) strategy in order to eliminate urban yellow fever outbreaks by 2026 (*World Health Organization, 2017*). The intermediate cycle currently only occurs in Africa when tree-hole breeding anthropophilic Aedes reach particularly high densities (*Monath and Vasconcelos, 2015*).

Control of yellow fever is primarily through vaccination, and there is no specific anti-viral treatment available. The 17D vaccine is live attenuated and was developed in 1936 (*Monath, 2005*). The vaccine is considered safe; estimates of adverse event occurrence are at most 0.6 per 100,000 doses, and reactions are generally mild (*de Menezes Martins et al., 2015*). Immunity due to yellow fever vaccination is suggested to be lifelong with WHO recommendations recently updated to reflect this (*Staples et al., 2015*). Efficacy is also thought to be high with recent estimates suggesting that serological response was 97.9% (95% credible interval [CrI] [82.9–99.7]) (*Jean et al., 2016*). An issue with the vaccine is production and corresponding stockpiles. As the vaccine is live, production is slow, which has led to vaccine shortages for large outbreaks and for travellers (*Gershman et al., 2017*). As a result of this, fractional dosing has become a recommendation in outbreak settings (*Barrett, 2020*).

Due to limited vaccine supply, efficient planning of interventions is vital to avoid large outbreaks. To facilitate this, robust estimates of disease burden and projections of future dynamics are key. Previous studies have focused on evaluating historical vaccine impact and projecting future potential impact. *Garske et al., 2014* produced vaccine impact estimates for the African endemic region focusing on mass vaccination campaigns until 2013. They found that mass vaccination activities have averted burden by 57% (95% CI [54–59]) in countries where they took place, accounting for 27% (95% CI [22–31]) of the burden across the region. More recently, *Shearer et al., 2018* examined the impact of vaccination globally, across Africa and South America, and found that (all) vaccination activities averted between approximately 94,0000 and 119,000 cases each year.

In this study, we refine and extend the model of Garske et al. to encompass new geographic regions (South America) and new data (on occurrence, serology, vaccination, and NHPs), and produce updated estimates of burden and vaccine impact for yellow fever. In the following sections, we describe the new data and extension to the modelling approach, particularly focusing on the updated model of yellow fever occurrence. Then we present results of our projected transmission intensity considering uncertainty from estimation and the structural uncertainty of the models. Finally, we present burden estimates and reassess the impact of mass vaccination activities in Africa.

## Materials and methods

We expand the existing framework of *Garske et al., 2014* for Africa to account for transmission in South America as well. Countries are included in the analysis if they have been listed as at risk, endemic, or potentially at risk for yellow fever (*World Health Organization, 2017*). We fit a generalised linear model (GLM) of yellow fever reports to occurrence data available from 1984 to 2019, shown in *Figure 1*. This occurrence data denotes whether yellow fever has been reported at all over the observation period, irrespective of number of cases. The GLM then provides a probability of a reported yellow fever outbreak for the entire region. In order to estimate the force of infection that would result in these outbreaks, we use serological survey data. We use this to independently estimate the seroprevalence in the survey locations and thus the force of infection. These individual estimates allow us to calculate a probability of detection over the observation period which we may then use to provide force of infection estimates for the entire region. Finally, using demographic and vaccination data, we can calculate the burden in all provinces.

### Data

We combine multiple data sets within a Bayesian framework to account for areas with sparse data and under-reporting. The model is estimated at province level, to match the available occurrence data. All data was from secondary sources, and ethics approval was thereby not required. *Figure 2* summarises the included data.

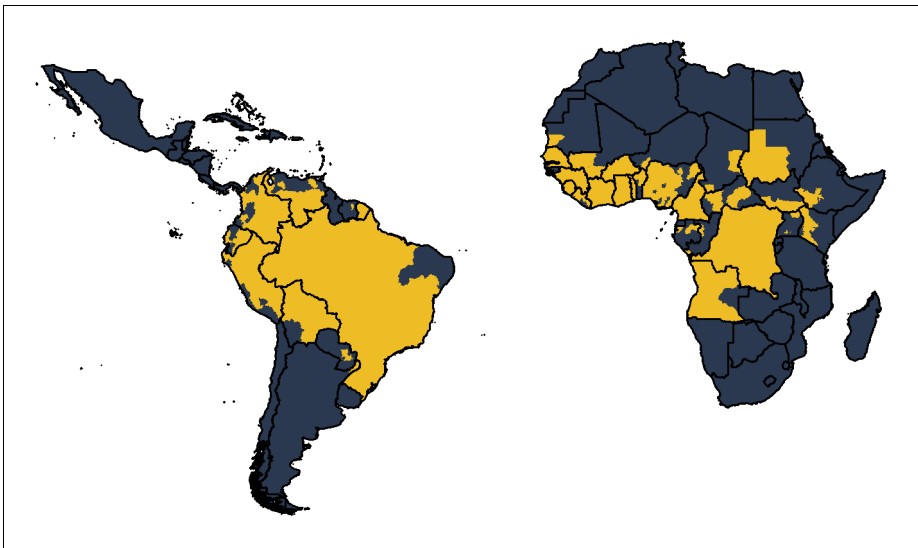

**Figure 1.** Global occurrence of yellow fever at province level. Occurrence since 1984 is shown in yellow.

## Global yellow fever occurrence

A database of yellow fever occurrence was collated. This was compiled in two parts: occurrence in Africa was compiled originally in Garske et al. and has been subsequently maintained and updated (*Garske et al., 2014*; *Gaythorpe et al., 2019*; *Jean et al., 2020*). Occurrence of yellow fever in South America was collated by *Hamlet et al., 2019*. Reports of yellow fever in humans were assembled for both continents from sources including the weekly epidemiological record (*World Health Organization, 2009*), disease outbreak news (*World Health Organization, 1996*), WHO yellow fever surveillance database (YFSD), Brazilian Ministry of Health, and Pan-American Health Organisation (PAHO). The outbreak dataset for Africa up to 2018 is available to download from: https:// github.com/kjean/YF_outbreak_PMVC/tree/main/formatted_data (*Gaythorpe, 2021*; copy archived at swh:1:rev:14703d7c5c7f63df6de04b81d5a48751604a906a). The cases of yellow fever were included if they were lab-confirmed through polymerase chain reaction. The YFSD includes confirmed and suspected yellow fever occurrence. Due to the low proportion of suspected cases in the database being due to yellow fever, this is used as a measure of surveillance effort where the incidence of suspected cases is aggregated to country level and divided by population to be used as a covariate in the GLM, following *Garske et al., 2014*.

## Vaccination coverage and demography

We use the methodology of Hamlet et al. and Garske et al. with updated data sets and additional data for South America in order to estimate vaccination coverage across the regions (*Garske et al., 2014*; *Hamlet et al., 2018*). The coverage estimates using this methodology are visualised and available to download at district level from 1940 to 2050 in the POLICI shiny application (*Hamlet et al., 2018*). The coverage is informed by historic data on mass-vaccination activities, reactive campaigns, recent preventive mass vaccination campaigns, and routine infant vaccination (*Durieux, 1956*; *Moreau et al., 1999*; *World Health Organization/ UNICEF, 2015*). Further data was provided by the Ministry of Health for Brazil.

Demographic data was obtained from the United Nations World Population Prospects (UNWPP), which provides country-level population sizes (*World population prospects, 2017*). These were disaggregated to province level using Landscan data on population distributions (*Dobson et al., 2000*; *LandScan, 2017*). Age distributions were assumed to be the same across all provinces, and population distributions were assumed not to substantially vary over the observation period. Landscan provides population size estimates at 1/120 degree resolution. Combining this with UNWPP, we arrive at the total number of individuals in each age group and province over time.

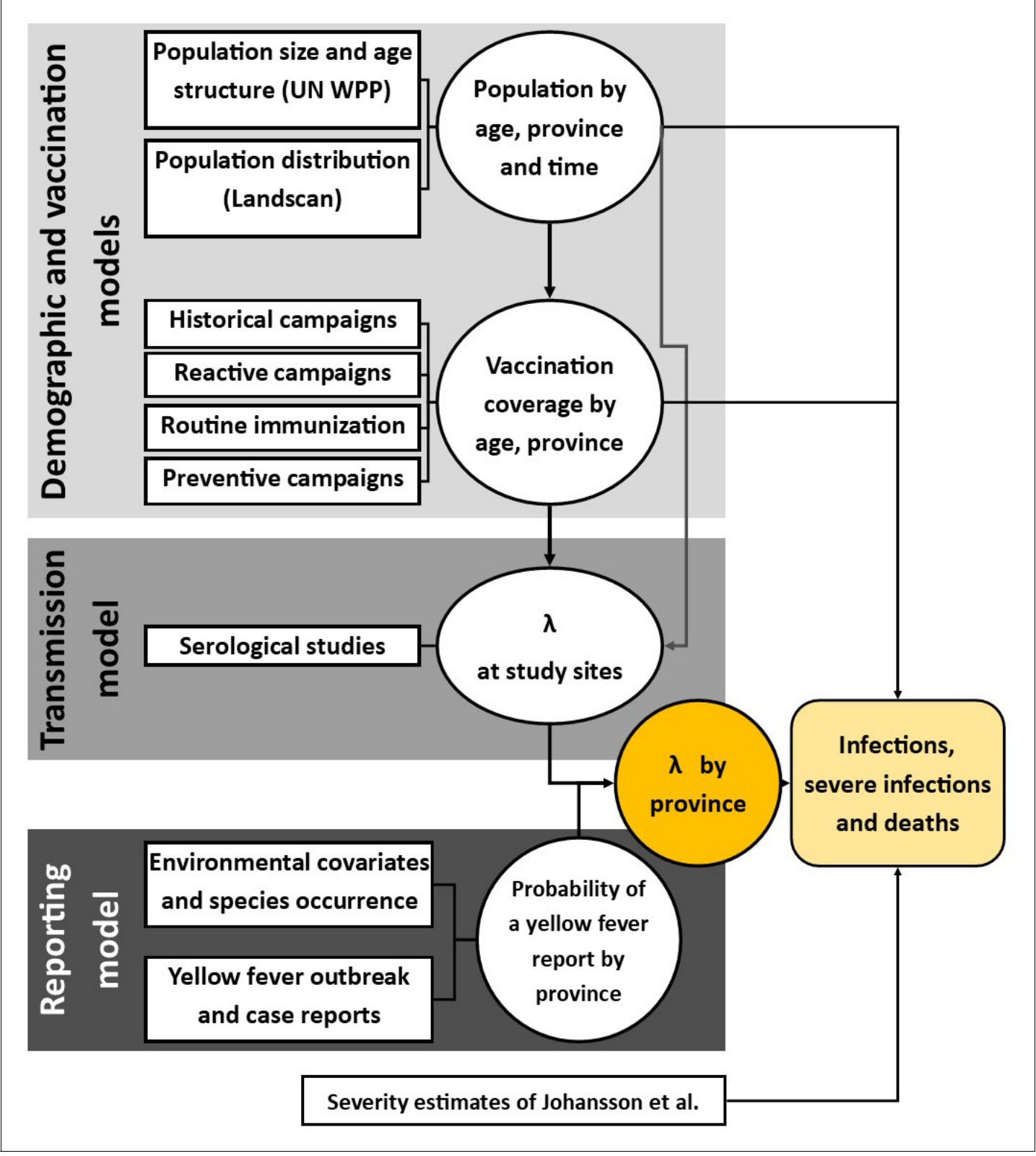

**Figure 2.** Diagram of models and data sources where $\lambda$ denotes the force of infection. Circles denote a product of calculation or inference; square boxes denote data sources. Adapted from *Gaythorpe et al., 2019*.

## Environmental and species occurrence data

The GLM of yellow fever occurrence was developed to account for dependence on environmental conditions, habitat suitability, and occurrence of the NHPs. Covariates include measures of enhanced vegetation index, altitude, temperature, precipitation, and land cover types as well as NHP species occurrence, *Ae. aegypti* and *Ae. albopictus* occurrence , and *Ae. aegypti* temperature suitability (*Fick and Hijmans, 2017*; *NASA, L. D, 2001*; *Xie and Arkin, 1996*; *Kraemer et al., 2015*).

Covariate data sets were available as gridded datasets of various spatial resolution. These were aggregated to province level, the same scale as the occurrence data. Temperature, altitude, and precipitation data was obtained from Worldclim version 2.0 (*Fick and Hijmans, 2017*). These were aggregated by calculating the mean, maximum, or minimum over the area of the province. Land cover was obtained from MODIS (*Friedl and Sulla-Menashe, 2019*; *Friedl and Sulla-Menashe, 2015*). This was aggregated by examining the proportion of each province coverage by a land cover type. NHP species distributions were obtained from the IUCN red list (*IUCN, 2019*). Occurrence of *Ae. aegypti* and albopictus was obtained from the supplementary information of *Kraemer et al., 2015*.

Prior to fitting, all variables were scaled to unit variance.

## Serological surveys

We use serological surveys to assess transmission intensity in specific regions. Unfortunately, these are only available in the African endemic region. We include all surveys included in Gaythorpe et al. as well as a newly published survey undertaken in Kenya (*Gaythorpe et al., 2019*; *Chepkorir et al., 2019*; *Diallo, 2010*; *Kuniholm et al., 2006*; *Merlin et al., 1986*; *Omilabu et al., 1990*; *Tsai et al., 1987*; *Werner et al., 1984*). In the majority of surveys, individuals known to have been vaccinated are omitted; however, in south Cameroon, this information is unavailable and so we estimate an additional vaccination factor. In the study of Chepkorir et al., we include vaccinated proportions as stated in their evaluation after omitting those with unknown status. Where it is possible to determine whether there had been outbreaks affecting the survey, surveys describing outbreak seroprevalence were omitted.

## Covariates for GLM

A full list of covariates is provided below; these are aggregated per province:

1. Annual temperature maximum, minimum, and range calculated from Worldclim (*Fick and Hijmans, 2017*),
2. Population size (on a log scale) from UNWPP disaggregated using Landscan (*World population prospects, 2017*; *Dobson et al., 2000*; *LandScan, 2017*),
3. Annual precipitation maximum, minimum, and mean calculated from Worldclim (*Fick and Hijmans, 2017*),
4. Enhanced vegetation index maximum, minimum, range, and mean calculated from NASA's Land Processes Distributed Active Archive Center data (*NASA, L. D, 2001*),
5. Middle infrared reflectance maximum, minimum, and mean calculated from NASA's Land Processes Distributed Active Archive Center data (*NASA, L. D, 2001*),
6. Proportion of land cover types such as savanna or grasslands were obtained from MODIS (*Friedl and Sulla-Menashe, 2019*; *Friedl and Sulla-Menashe, 2015*),
7. Occurrence of *Ae. aegypti* and *Ae. albopictus* as provided in the supplementary data of *Kraemer et al., 2015*, both vectors are included as both can carry yellow fever, although it is worth noting that *Ae. aegypti* is the main urban vector,
8. Occurrence of all NHP families such as cercopithecidae from the IUCN redlist (*IUCN, 2019*),
9. Mean altitude per province calculated from Worldclim (*Fick and Hijmans, 2017*),
10. Temperature suitability index for *Ae. aegypti* as described in *Gaythorpe et al., 2020*.

Further details on how the NHP data was aggregated and how the temperature suitability was calculated are provided below.

## Non-human primates

NHP data was acquired from the IUCN redlist (*IUCN, 2019*). This provided presence range maps at species level as polygons. NHP species whose range polygon covered more than 10% of a province were considered to be present in that province. In order to produce maps of primate species

richness, we perform a count of all species belonging to a family in each province. We include all NHP primate families in the covariate selection process. If a primate family is included in the resulting model, the primate species richness is classed as the number of primate species in that family that are present in the province.

## Temperature suitability

Temperature suitability was calculated as in *Gaythorpe et al., 2020*. The form of the temperature suitability index is given by:

$$z(T) = \frac{a(T)^2 \exp(-\mu(T)\rho(T))}{\mu(T)}$$

where the bite rate, extrinsic incubation period, and mosquito mortality, given by $a, \rho$, and µ, respectively, are affected by temperature $T$ in the following ways:

$$\begin{aligned} a(T) &= a_c T(T - a_{T_0})(a_{Tm} - T)^{0.5} \\ \rho(T) &= 1/(\rho_c T(T - \rho_{T_0})(\rho_{Tm} - T)^{0.5}) \\ \mu(T) &= 1/(-\mu_c(T - \mu_{T_0})(\mu_{T_m} - T)) \end{aligned}$$

following (*Mordecai et al., 2017*). The subscripts $c, 0$, and $m$ represent the positive rate constant, minimum temperature, and maximum temperature for each thermal response model. These were estimated within a Bayesian framework, and we retain the point estimates shown in Table 2 (*Gaythorpe et al., 2020*).

## Models

We extend the model of *Garske et al., 2014* to account for yellow fever burden in South America as well as Africa. We update currently included data and include further data where necessary to expand the scope of the modelling.

## Seroprevalence

We assume a constant force of infection for each province over the observation period. This is the same as Garske et al. and was found to be a better reflection of available data than that another, dynamic, model variant (*Gaythorpe et al., 2019*). We assume homogeneous mixing and account for vaccination using the following form for $s(\lambda, u)$, the expected seroprevalence in age group $u$ given force of infection $\lambda$:

$$s(\lambda, u) = 1 - (1 - \frac{\sum_{a \in u}(1 - exp(-\lambda a)p_a)}{\sum_{a \in u}(p_a)})(1 - \frac{\sum_{a \in u} v_a p_a}{\sum_{a \in u} p_a}),$$

where $a$ indexes the annual age groups, $p_a$ is the population age distribution, and $v_a$ is the vaccination coverage in age group $a$. The binomial log likelihood is then given by the following:

$$\log L_{sero} = \sum_u \log N_u K_u s(\lambda, u)^{K_u} (1 - s(\lambda, u)^{N_u}),$$

where $N_u$ is the number of samples in age group $u$ and $K_u$ is the number of positive samples in age group $u$.

## GLM of the presence/absence of yellow fever reports

A GLM was fitted to the data set of yellow fever occurrence from 1984 to 2019 at province level. The data is assumed to be binomially distributed and a complementary log–log link function is used such that model predictions in province $i$, $q_i$, are given by

$$q = 1 - exp(-e^{X\beta}),$$

where $X$ denotes the matrix of covariates and $\beta$ indicates the parameter vector to be fitted. The log-likelihood is given by

$$\log L_{glm} = \sum_i (y_i \log(q_i) + (1 - y_i) \log(1 - q_i)),$$

where $y_i$ denotes the presence/ absence in province $i$.

The occurrence of yellow fever depends on a number of environmental factors as well as the abundance and distribution of the vector and NHP hosts. We consider many of these variables as potential covariates in the model. As with Garske et al., the number of covariates to consider is large and has been extended for the current work by the inclusion of NHPs and temperature suitability. As such we perform a selection process, detailed below:

1. We remove covariates that are not significantly associated with the data. For each covariate, we fit a univariate GLM to the data using the base R function, glm. We remove covariates with p-value<0.1; in this case, all covariates were significant.
2. Highly correlated covariates are clustered such that the pairwise correlation in each cluster exceeds 0.75. This produces 38 clusters.
3. We choose one covariate from each cluster to be further examined. Here, the covariate with the maximum absolute correlation with the data is chosen.
4. The function stepAIC from the MASS package is used to further whittle the list of covariates down (*Venables and Ripley, 2002*). We choose the multiplier of the number of degrees of freedom such that the test criterion is BIC: the Bayesian information criterion instead of AIC: the Akaike information criterion.
5. The final step is to use the R package, bestglm to produce the best 20 models according to BIC (*McLeod and Xu, 2018*). This uses the complete enumeration algorithm.

All models then included a measure of surveillance quality. For the 21 countries within the yellow fever surveillance database, specific data on reporting per capita was available. For countries not covered by the yellow fever surveillance database, and thus without an independent estimate of surveillance, individual country factors were fitted. However, countries not considered at risk were grouped together in order to have one country factor. This is in order to avoid infinite parameter estimates in areas which are known not to have yellow fever reports.

## Transmission intensity

The transmission intensity estimates arising from the serology allow us to calculate the number of infections over the observation period in the areas where surveys were conducted. We link this to the probability of yellow fever report through a Poisson reporting process with a probability of the detection. This is calculated by comparing the GLM to predictions of the seroprevalence models in the following way:

$$q_i = 1 - (1 - \rho_i)^{n_{inf,i}},$$

where $\rho_c$ is the per-country probability of detection, $q_i$ is the probability of a report in province $i$, provided by the GLM, and $n_{inf,i}$ is the number of infections in province $i$, provided by the seroprevalence model. This means that the probability of detection can be linked to the GLM covariates by:

$$n_{inf,i} \log(1 - \rho_c) = \exp(X\beta),$$

and in terms of the country factors, GLM covariates $\beta_c$, and $b$, the baseline surveillance quality by:

$$\log(-\log(1 - \rho_c)) = \beta_c + b.$$

Once the probability of detection has been estimated for each province with a serological study, we take the mean over each and use the resulting probability to extrapolate transmission intensity in areas where there are currently no seroprevalence studies.

## Estimation

The best-fitting models, according to BIC, were estimated within a Bayesian framework described in *Gaythorpe et al., 2019* including code used. The estimation is divided into two phases. The GLMs are estimated using adaptive Markov chain Monte Carlo (MCMC) sampling, whereas the seroprevalence models are estimated within a product space framework with the probability of the force of

infection model set to 1; as such, the estimation becomes an adaptive MCMC with log-transformed parameters.

Prior distributions were chosen in many cases to match those of Garske et al. Country factors retain the Gaussian prior distribution with mean 0 and standard deviation 2 except for the countries considered low risk, whose country factor had truncated normal prior with mean 0 and standard deviation 30 and limits $[0, \infty]$, designed to be uninformative and positive. The same prior was used for the GLM coefficient for the aggregated NHP species richness and the converse for the GLM coefficients for temperature range and altitude which were assumed to be negative. All other GLM coefficient priors were normal with mean 0 and standard deviation 30 to be uninformative. The force of infections for each seroprevalence study had exponential priors with rate parameter 0.001, although this was varied in earlier estimation, and the vaccine efficacy had truncated normal prior with mean 0.975 and standard deviation 0.05, according to *Jean et al., 2016*; this was truncated to [0,1].

## Ensemble predictions

We propagate uncertainty from both the parameter estimation and model structure. This is done through sampling proportionally from the posterior distributions of all 20 of the best-fitting models to produce 500 force of infection and thus burden predictions. We sample proportional to the area under the curve (AUC) of each model fit. We also considered sampling proportional to likelihood; however, the AUC was used to compare our models with previous works such as *Garske et al., 2014* and was more computationally efficient for larger sample sizes. In order to produce estimates of the number of severe infections and deaths per province and year, we scale the model output, infections, by sampling from the full uncertainty ranges of *Johansson et al., 2014* for the proportion of infections considered severe and, of those, the proportion that will then die of yellow fever.

Whilst we propagate structural and parameter uncertainty to the predictions from the estimation of our models, the models themselves are inherently deterministic. As such, we will not capture the potential for large outbreaks, or stochastic die-out where a spillover event fails to spark an outbreak. Because of this, the burden estimates we present should be considered as an average behaviour of the *potential* number of deaths caused by yellow fever given the vaccination coverage estimated. This means that whilst the projected burden over time will reflect the data used, the year-on-year variation may be missed.

All analyses, estimation, and the original draft of the manuscript were performed in R version 3.5.1.

## Role of funding source

This work was carried out as part of the Vaccine Impact Modelling Consortium, which is funded by Gavi, the Vaccine Alliance and the Bill and Melinda Gates Foundation. The views expressed are those of the authors and not necessarily those of the Consortium or its funders. The final decision on the content of the publication was taken by the authors. We acknowledge joint Centre funding from the UK Medical Research Council and Department for International Development. The funders had no role in study design, data collection and interpretation, or the decision to submit the work for publication.

# Results

## Regression model fitting and variable selection

All of the models included log (surveillance quality) and country factors for which country-specific surveillance information was not available. A total of 50 covariates were considered, all of which were significantly related to the data with threshold p=0.1. These were clustered into 39 groups leading to approximately $2.03 \times 10^{46}$ model permutations. Following the use of a step function based on BIC, we reduce this number to 13 covariates and retain the 20 best models including these, shown in *Table 1*.

Similar to Garske et al., all of the 20 best-fitting models included the log of population size, relating the probability of a report with the human population. All 20 models also included the temperature suitability at mean temperature which will limit the models in areas where the temperatures are

too extreme to sustain vector transmission dynamics; *Table 2* shows the parameters for the temperature suitability model. The species richness of three NHP families were included in all variants. These were aggregated in order to balance the contribution of the NHP host with the competence of vectors and human dynamics and populations. All model covariates are shown in *Figure 3*. Covariates such as *Ae. aegypti* occurrence, temperature range, altitude and barren, cropland, shrubland, and water body land cover were only included in some of the best models.

The 20 best-fitting models were also fit with full MCMC and the AUC calculated. These are shown in *Figure 4*. The AUCs of model variants are very similar with variant six generally the best. All model variants exceed the AUC of *Gaythorpe et al., 2019* which had point estimate of 0.916.

## Yellow fever occurrence

We predict yellow fever occurrence over the observation period in *Figure 5*. These ensemble predictions indicate high probabilities of report in the Amazon region of Brazil and West Africa. Note the results shown also include a measure of surveillance effort emphasising countries such as Angola.

## Seroprevalence

The model prediction captured the wide range of transmission intensity, see *Figure 6*. However, in certain conditions such as in Kenya area 1, the fit is affected by uncertainty concerning the vaccination status of included individuals. These results show similar qualitative fits to *Gaythorpe et al., 2019*. Indeed, there are only two additional included studies, those found in Kenya of *Chepkorir et al., 2019*. In some cases, some of the data points lie outside the 95% CrI, these generally align with areas where there is uncertainty in the vaccination status of included individuals or where it is indicative of an outbreak.

## Transmission intensity

In *Figure 7*, we show the median ensemble predictions of transmission intensity. In comparison to Garske et al., the force of infection in West Africa is slightly lower, and provinces in the Democratic Republic of the Congo (DRC) are highlighted as areas of high transmission intensity. However, the main area highlighted is that of Amazon in Brazil. See *Figure 7—figure supplement 1* for the coefficient of variation between 100 samples from each of the 20 models, sampled equally; this further highlights areas of low transmission intensity such as the Sahara having higher degrees of uncertainty.

## Burden

The annual potential number of deaths and severe infections are shown in *Table 3* with deaths per country in 2018 given in *Figure 8*. We estimate that in 2018 there were approximately 109,000 (95% CrI [67,000–173,000]) severe infections and 51,000 (95% CrI [31,000–82,000]) deaths due to yellow fever in these two regions. Burden is distributed unevenly between countries and continents. The highest burden is seen in the DRC due to a high force of infection and low vaccination coverage. In contrast, Brazil sees the fourth highest burden purely due to high force of infection in the Amazon region rather than low vaccination coverage. The majority of the burden occurs in Africa, which holds for all years shown.

## Impact of mass vaccination campaigns in Africa

It has been shown that mass vaccination campaigns can produce long-lasting effects on disease burden. We examine the effects of mass vaccination activities from 2006 until 2019 in countries in Africa, to continue the analysis of *Garske et al., 2014*, in *Figure 9* and in the figure supplement for 2013. We find large reductions in all countries with mass vaccination campaigns. In 2018, the largest reductions are of approximately 73% (95% CrI [64–79]) in Benin, 73% (95% CrI [60–81]) in Togo, and 61% (95% CrI [52–68]) in Liberia. This demonstrates the continued benefit of those campaigns.

Overall, the reductions in the number of deaths per year are substantial, shown in *Table 4*. These amount to approximately 10,000 (95% CrI [6,000–17,000]) deaths averted in 2018 due to mass vaccination activities in Africa, corresponding to a 47% reduction (95% CrI [10–77]) in deaths.

**Table 1.** Composition of the 20 best-fitting generalised linear models of yellow fever reports.
Surveillance quality is also included in all models. If an entry is 1, that covariate is included, if an entry is 0, that covariate is not included. Abbreviations used: MIR = middle infrared reflectance, Temp. = temperature., occ. = occurrence.

| Model | Cercopithecidae occ. | Cebidae occ. | Population (log) | Temp. suitability (mean) | Grasslands | Savanna | Evergreen broadleaf forests | Ae. aegypti occ. | Aotidae occ. | Woody savanna | Temp. range | Maximum MIR | Altitude | BIC |
|---|---|---|---|---|---|---|---|---|---|---|---|---|---|---|
| 1 | 1 | 1 | 1 | 1 | 1 | 0 | 0 | 1 | 1 | 0 | 0 | 1 | 0 | 870 |
| 2 | 1 | 1 | 1 | 1 | 1 | 1 | 1 | 1 | 1 | 1 | 1 | 1 | 1 | 872 |
| 3 | 1 | 1 | 1 | 1 | 1 | 0 | 0 | 1 | 1 | 0 | 0 | 1 | 1 | 872 |
| 4 | 1 | 1 | 1 | 1 | 1 | 1 | 1 | 1 | 1 | 1 | 1 | 0 | 0 | 872 |
| 5 | 1 | 1 | 1 | 1 | 1 | 1 | 1 | 1 | 1 | 1 | 1 | 0 | 1 | 873 |
| 6 | 1 | 1 | 1 | 1 | 1 | 1 | 1 | 1 | 1 | 0 | 0 | 1 | 0 | 873 |
| 7 | 1 | 1 | 1 | 1 | 1 | 1 | 1 | 1 | 1 | 1 | 1 | 1 | 0 | 873 |
| 8 | 1 | 1 | 1 | 1 | 1 | 1 | 1 | 1 | 1 | 0 | 0 | 1 | 1 | 873 |
| 9 | 1 | 1 | 1 | 1 | 1 | 1 | 0 | 1 | 1 | 0 | 0 | 1 | 0 | 873 |
| 10 | 1 | 1 | 1 | 1 | 1 | 1 | 1 | 1 | 1 | 1 | 0 | 1 | 1 | 874 |
| 11 | 1 | 1 | 1 | 1 | 1 | 1 | 1 | 1 | 1 | 0 | 1 | 1 | 0 | 874 |
| 12 | 1 | 1 | 1 | 1 | 1 | 1 | 1 | 1 | 1 | 0 | 1 | 1 | 1 | 874 |
| 13 | 1 | 1 | 1 | 1 | 1 | 1 | 1 | 1 | 1 | 1 | 0 | 1 | 0 | 875 |
| 14 | 1 | 1 | 1 | 1 | 1 | 1 | 1 | 0 | 1 | 0 | 0 | 1 | 0 | 875 |
| 15 | 1 | 1 | 1 | 1 | 1 | 0 | 0 | 0 | 1 | 0 | 0 | 1 | 0 | 875 |
| 16 | 1 | 1 | 1 | 1 | 1 | 0 | 0 | 1 | 1 | 1 | 0 | 1 | 0 | 875 |
| 17 | 1 | 1 | 1 | 1 | 1 | 1 | 1 | 1 | 1 | 0 | 1 | 0 | 0 | 875 |
| 18 | 1 | 1 | 1 | 1 | 1 | 0 | 0 | 1 | 1 | 0 | 1 | 1 | 0 | 875 |
| 19 | 1 | 1 | 1 | 1 | 1 | 1 | 0 | 1 | 1 | 0 | 0 | 1 | 1 | 875 |
| 20 | 1 | 1 | 1 | 1 | 1 | 1 | 1 | 0 | 1 | 1 | 1 | 0 | 0 | 876 |

## Discussion

In this study, we further developed models of yellow fever transmission in Africa and South America. We calculated disease burden in terms of severe infections (or cases) and deaths from an ensemble of best-fitting GLMs of yellow fever reports between 1984 and 2019 coupled to catalytic models of seroprevalence. We used this approach to evaluate the impact of mass vaccination campaigns in Africa as well as produce updated burden estimates of yellow fever in endemic regions.

We estimate that there are between 63,000 and 158,000 severe infections of yellow fever in Africa, resulting in 29,000–75,000 deaths. In South America, we estimate there are 4000–15,000 severe infections, resulting in 2000–7000 deaths. These estimates are contained within the bounds of Garske et al. who estimated between 51,000 and 380,000 severe infections and between 19,000 and 180,000 deaths occur each year in Africa. We also compare to *Shearer et al., 2018* who found approximately 256,000 cases on the African continent and 28,000 within Latin America, which are at the higher end of our predicted ranges. All the above estimates fall within a similar range despite different scopes and modelling approaches.

In order to produce our burden estimates, we first estimate transmission intensity through a force of infection for each province. These estimates differ from those of Garske et al. in West Africa and the DRC. In order to account for the extended model scope, we revisited the covariates used in the GLM leading to the changes in West African forces of infection shown. This can partly be explained by the exclusion of certain covariates, such as longitude, and the inclusion of others, such as NHP species richness. The latter highlights the DRC as an area of high transmission potential, increasing the local estimates of force of infection. The former reduces the force of infection estimates in West Africa. This decrease in West Africa has led to our range of burden lying within the lower range of Garske et al., and whilst the same proportional impact of vaccination is found, the number of deaths averted is also in the lower range of previous estimates. These differences may also indicate a general sensitivity of the approach as, whilst burden is generally agreed to be higher in West Africa, it may be difficult to determine the exact magnitude from occurrence data alone. In South America, the force of infections are estimated to be highest in the Amazon region of Brazil, in part due to the high NHP species richness found there. This is consistent with vaccination efforts which have focused on this area leading to relatively low burden despite the high intensity of transmission.

In refining the GLMs of yellow fever occurrence, we expanded the pool of possible covariates. This was partly facilitated by new data becoming available, such as NHP species occurrence, and partly motivated by the need to capture more of the inherent variability of yellow fever occurrence, through a temperature suitability index. We find that certain features such as human population size and certain land cover types were consistently featured in the top models of occurrence. Similarly, primate families, Cercopithecidae, Cebidae, and Aotidae—were all found to be important to yellow fever occurrence. The collection of covariates and model choice lead to high AUCs for each model ranging from 0.935 to 0.949, higher than *Gaythorpe et al., 2019*. One element that is omitted explicitly is vector abundance although we do include occurrence of *Ae. aegypti* and *Ae. albopictus*. In recent years, there have been a number of excellent efforts to map and predict vector distributions (*Kraemer et al., 2015*; *Brady et al., 2014*). We utilise many of the same model covariates and as such have chosen to omit modelled vector distributions in this analysis. Additionally, yellow fever is transmitted by many vectors whose distributions have yet to be characterised.

There are a number of additional limitations with the range of covariates used. We utilise NHP species presence/absence data which we aggregate to province level through counts. As such, we do not have information of the population sizes of NHP, only diversity. This could be reassessed as further data becomes available. Additionally, as a component of the covariate selection process, we

---

**Table 2.** Temperature suitability index parameter values.
The subscripts $c$, 0, and $m$ represent the positive rate constant, minimum temperature, and maximum temperature for each thermal response model. Parameter $a$ corresponds to bite rate, $\rho$ corresponds to extrinsic incubation period, and $\mu$ corresponds to mosquito mortality.

|  | $a_c$ | $a_{T_0}$ | $a_{T_m}$ | $\rho_c$ | $\rho_{T_0}$ | $\rho_{T_m}$ | $\mu_c$ | $\mu_{T_0}$ | $\mu_{T_m}$ |
|---|---|---|---|---|---|---|---|---|---|
| Value | 2.72e-4 | 2.24 | 40.13 | −0.75 | 12.71 | 38.05 | 1.36e-4 | 17.33 | 42.20 |

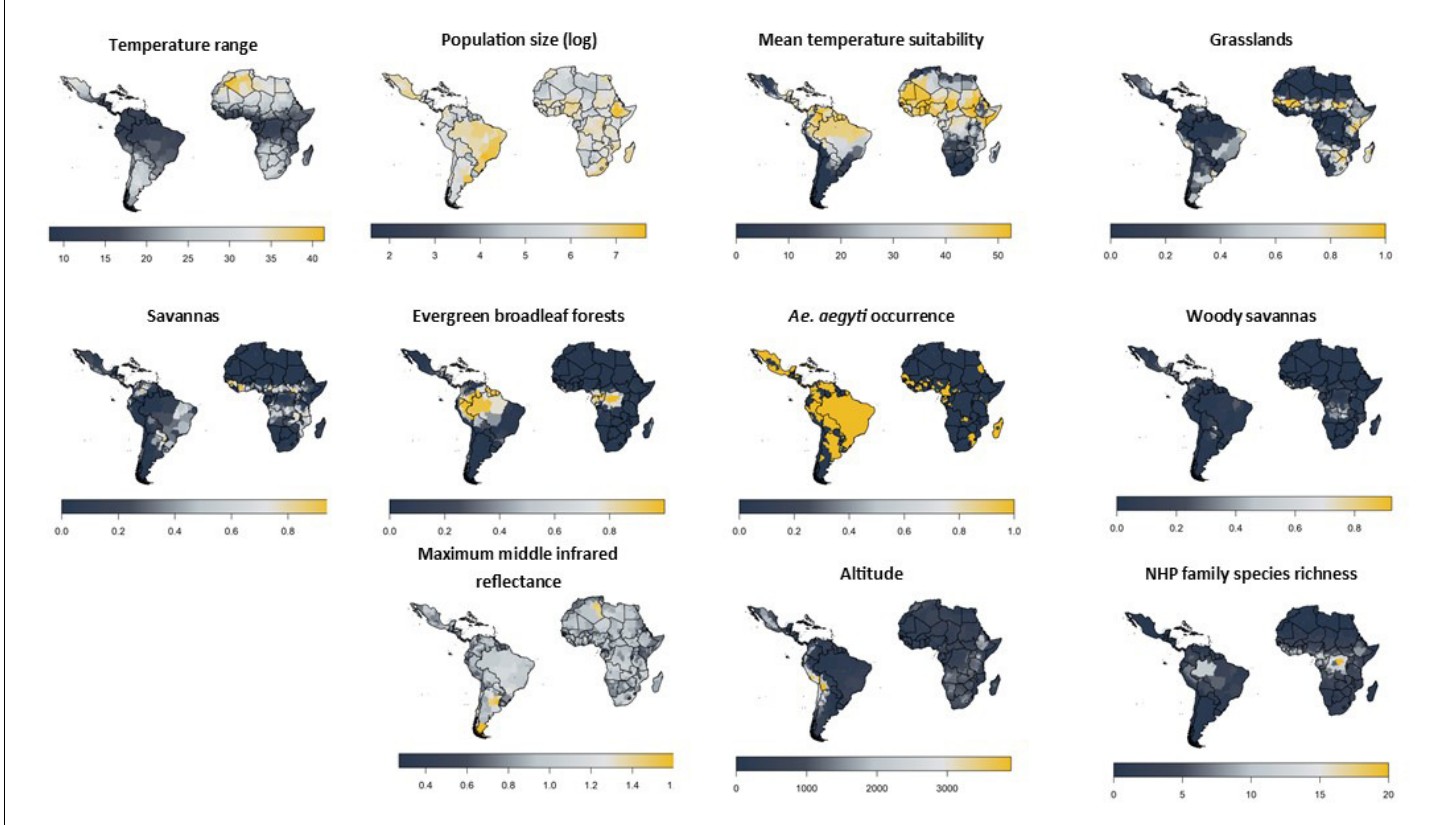

**Figure 3.** Included model covariates. Species richness is the sum of all NHP species present per province from families listed in *Table 1* and will vary as families are included/excluded. See *Figure 3—figure supplement 1–4* for trace plots of all parameters.

The online version of this article includes the following figure supplement(s) for figure 3:

**Figure supplement 1.** Trace plots from estimation of model variant 17 as an example of convergence.

**Figure supplement 2.** Trace plots from estimation of model variant 17 as an example of convergence.

**Figure supplement 3.** Trace plots from estimation of model variant 17 as an example of convergence.

**Figure supplement 4.** Trace plots from estimation of model variant 17 as an example of convergence.

cluster our covariates based on their correlation with each other. As such, NHP families who coexist in the same geographic locations are put in the same cluster, for example Atelidae, leading the model selection process to potentially include a NHP that is present in areas with yellow fever, but not necessarily causing or carrying the virus. As with the other covariates, the selection process implies correlation not causation between the covariate and the occurrence of yellow fever.

Apart from the NHP families, we utilise the same environmental covariates for each continent. Whilst this improves consistency, we use elements such as temperature suitability for *Ae. aegypti* as a proxy for the vectors of yellow fever and there are different species in each continent, which may differ in their own ways to *Ae. aegypti*. An additional issue is that we aggregate our environmental covariates to province level, reducing resolution and potentially biasing the results. In this study, we feel the sparsity of the data, long time window of interest, and general uncertainty in other features will eclipse biases introduced by the aggregation mechanism, but if this model were to be refined spatially, the bias from aggregation could be readdressed. Finally, whilst we include structural uncertainties from the GLM, we do not included uncertainty in the covariates themselves.

We estimate our models from two main sources of data that we have updated where possible. The occurrence data was expanded to account for additional years and locations from *Garske et al., 2014*; *Gaythorpe et al., 2019*; *Jean et al., 2020*; *Hamlet et al., 2019*. Yet there are assumptions and uncertainty that are inherent in this data. Firstly, the case definition relies on a vague symptom set, which is prone to mis-attribution and may vary by location, this will affect under-reporting. We accommodate variation between countries though country-specific reporting factors in the GLMs;

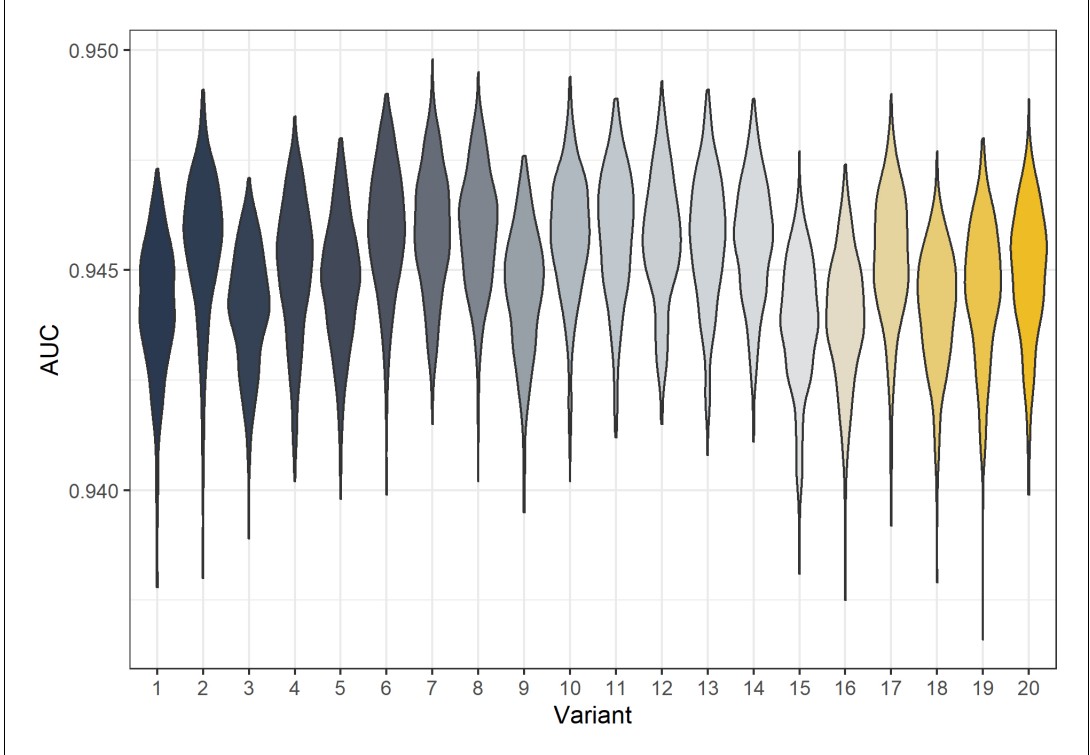

**Figure 4.** Posterior predicted area under the curve (AUC) for all model variants. The AUC are calculated for 500 samples from the posterior of each model variant.

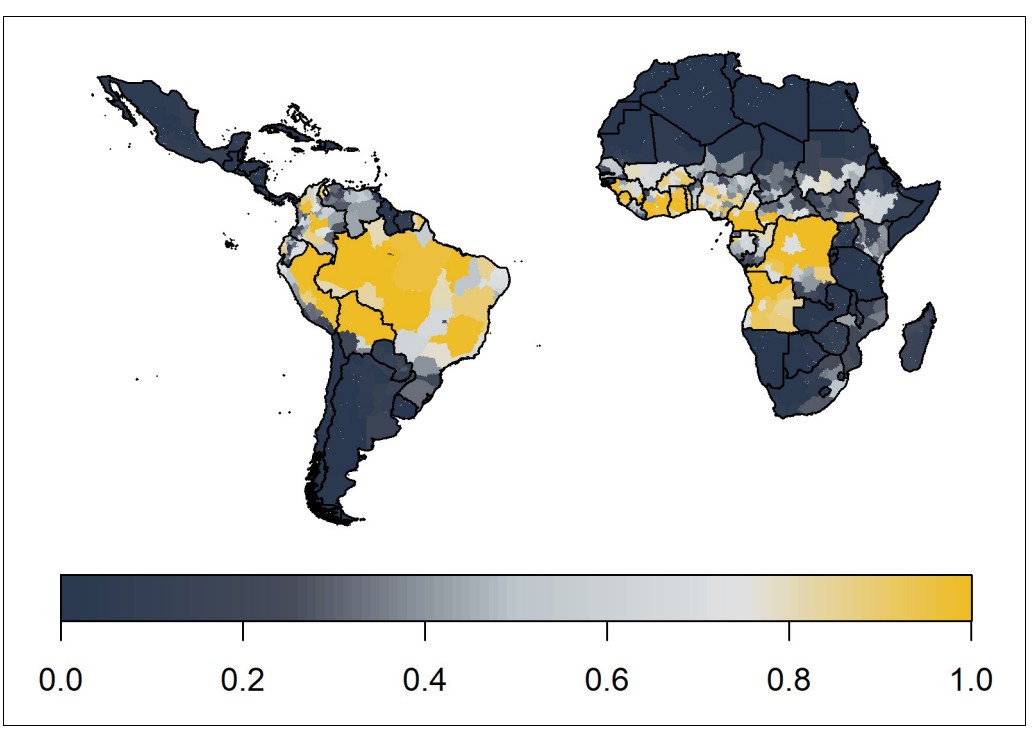

**Figure 5.** Median posterior predicted probability of a yellow fever report from ensemble predictions of the 20 best GLMs. This applies over the observation period 1984–2019.

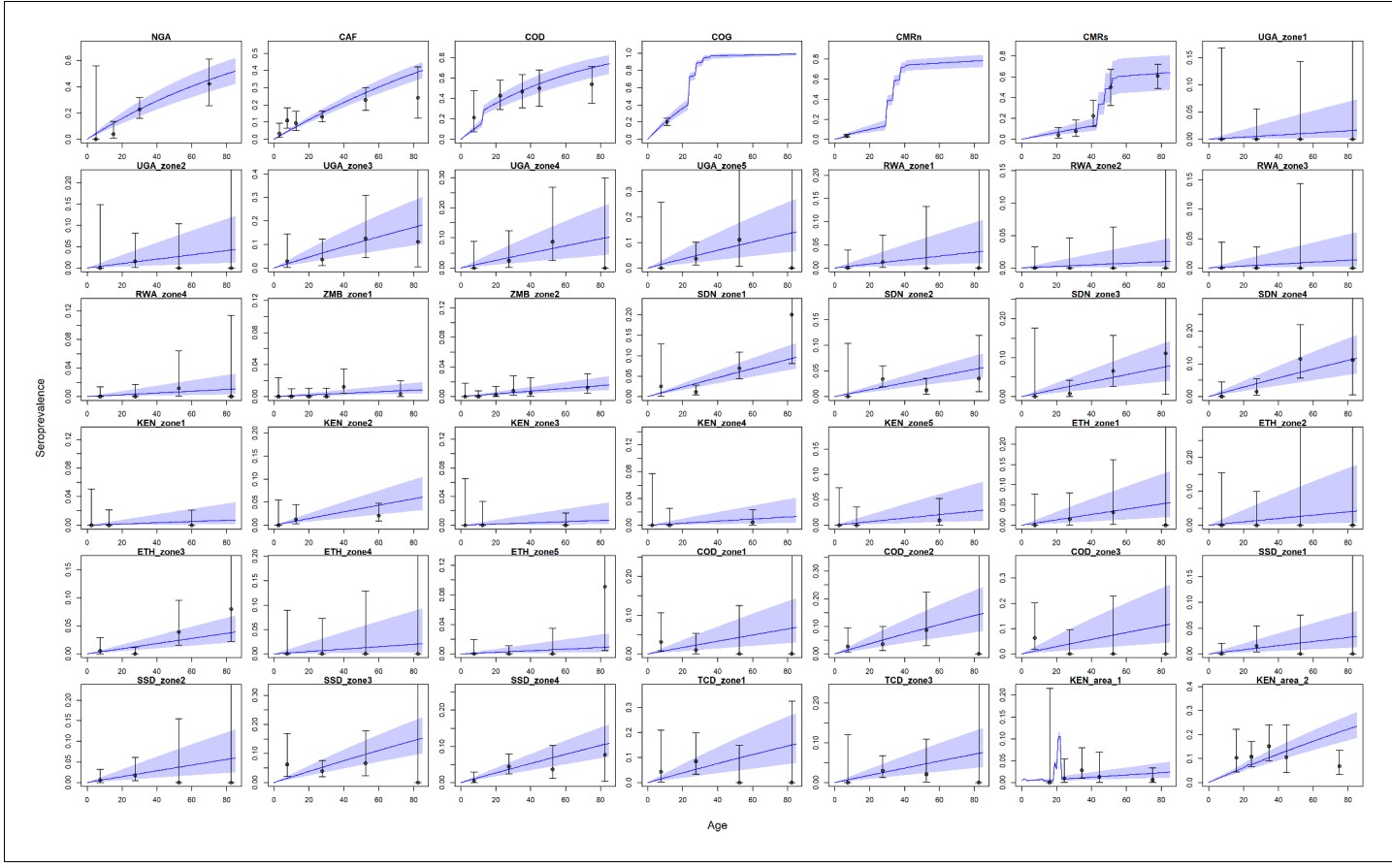

**Figure 6.** Seroprevalence predictions for each serological survey. Central blue line indicates median posterior predicted seroprevalence; blue area indicates 95% CrI. Dots indicate the data with error bar representing binomial confidence intervals. Countries are named by their ISO code with different ecological zones indexed 'zone x'. See *Figure 6—figure supplement 1* for posterior distribution of vaccine efficacy and vaccine factor for CMRs; see *Figure 6—figure supplement 2* for comparison of force of infection estimates under different prior distributions.

The online version of this article includes the following figure supplement(s) for figure 6:

**Figure supplement 1.** Prior and posterior distributions for vaccine efficacy and vaccine factor for CMRs.

**Figure supplement 2.** Comparison of force of infection estimates for the serological study sites using two prior formulations.

however, there are number of elements that contribute to surveillance which we essentially aggregate into one component. The most stark difference may be that surveillance in South America uses that the fact that some NHP species experience disease-related mortality as sentinels for yellow fever. In Africa, due to the co-evolution of NHPs and virus, NHPs are not known to be significantly affected in the same way. As such, the surveillance systems are substantially different.

One of the important components to assess inherent under-reporting of yellow fever is serology data. We have significantly expanded the data sources for this aspect of the model, with an addition of 36 studies compared to Garske et al. However, all studies are located in Africa, the high vaccination coverage in many of the provinces render conventional tests of seroprevalence uninformative in terms of assessing background infection. There are further issues that may arise in using seroprevalence. We take a positive serology test to indicate exposure to the disease but also protection, adhering to the conventional assumption that immunity to yellow fever is acquired after infection or vaccination and remains for a lifetime. However, there have been recent studies suggesting that this may not be the case in children. *Domingo et al., 2019* found that immunity against yellow fever waned in children following vaccination. If these results are representative of infant and child vaccination across the regions and time, our estimates of population immunity may need to be readdressed.

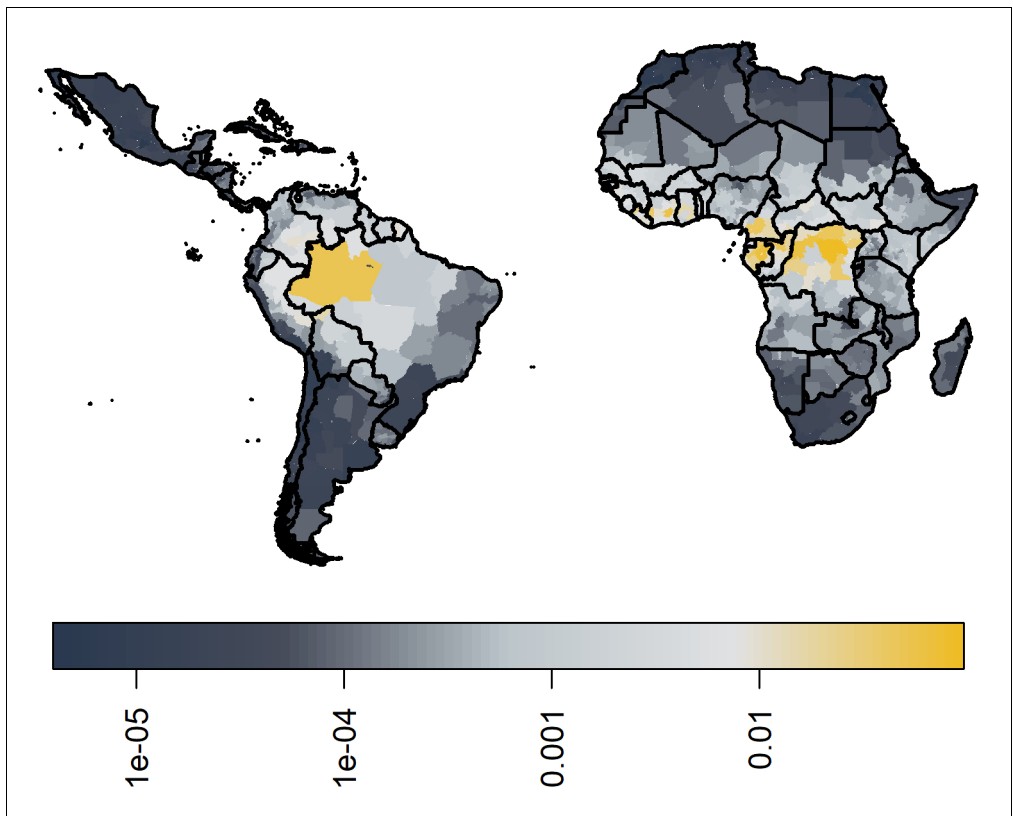

**Figure 7.** Median posterior predicted force of infection from ensemble predictions of the 20 best GLMs. Force of infections are assumed to be time invariant as such, these do not correspond to a particular year. See *Figure 6— figure supplement 1* for coefficient of variation.

The online version of this article includes the following figure supplement(s) for figure 7:

**Figure supplement 1.** Coefficient of variation in the force of infection estimates between 100 samples of each of the 20 best models.

Modelling yellow fever is inherently uncertain. We have attempted to quantify this uncertainty through a Bayesian framework and ensemble model predictions. However, there are still elements that we have not captured. Uncertainty in demography and vaccination are not propagated through our model results and yet both will be influential. Demography is captured from UNWPP and scaled according to Landscan. We assume relatively static age structures and, whilst UNWPP goes some way to accounting for population movements, we do not include them explicitly. Population movements not only effect the model directly but will also influence the resulting vaccination coverage

**Table 3.** Potential deaths and severe infections per year in Africa and South America from ensemble model projections.

| Continent | Year | Severe infections, median | Severe infections, 95% CrI low | Severe infections, 95% CrI high | Deaths, median | Deaths, 95% CrI low | Deaths, 95% CrI high |
|---|---|---|---|---|---|---|---|
| Africa | 1995 | 102,972 | 62,162 | 160,700 | 48,474 | 28,672 | 76,998 |
| Africa | 2005 | 122,101 | 74,915 | 192,773 | 57,182 | 34,446 | 90,736 |
| Africa | 2013 | 98,148 | 62,083 | 150,953 | 45,973 | 28,680 | 72,380 |
| Africa | 2018 | 100,952 | 63,001 | 158,362 | 47,318 | 29,162 | 74,981 |
| Americas | 1995 | 14,349 | 6528 | 26,016 | 6652 | 3026 | 12,577 |
| Americas | 2005 | 10,254 | 4988 | 18,436 | 4827 | 2265 | 8779 |
| Americas | 2013 | 8559 | 4264 | 15,043 | 3999 | 1969 | 7162 |
| Americas | 2018 | 8331 | 4306 | 14,608 | 3883 | 1971 | 7033 |

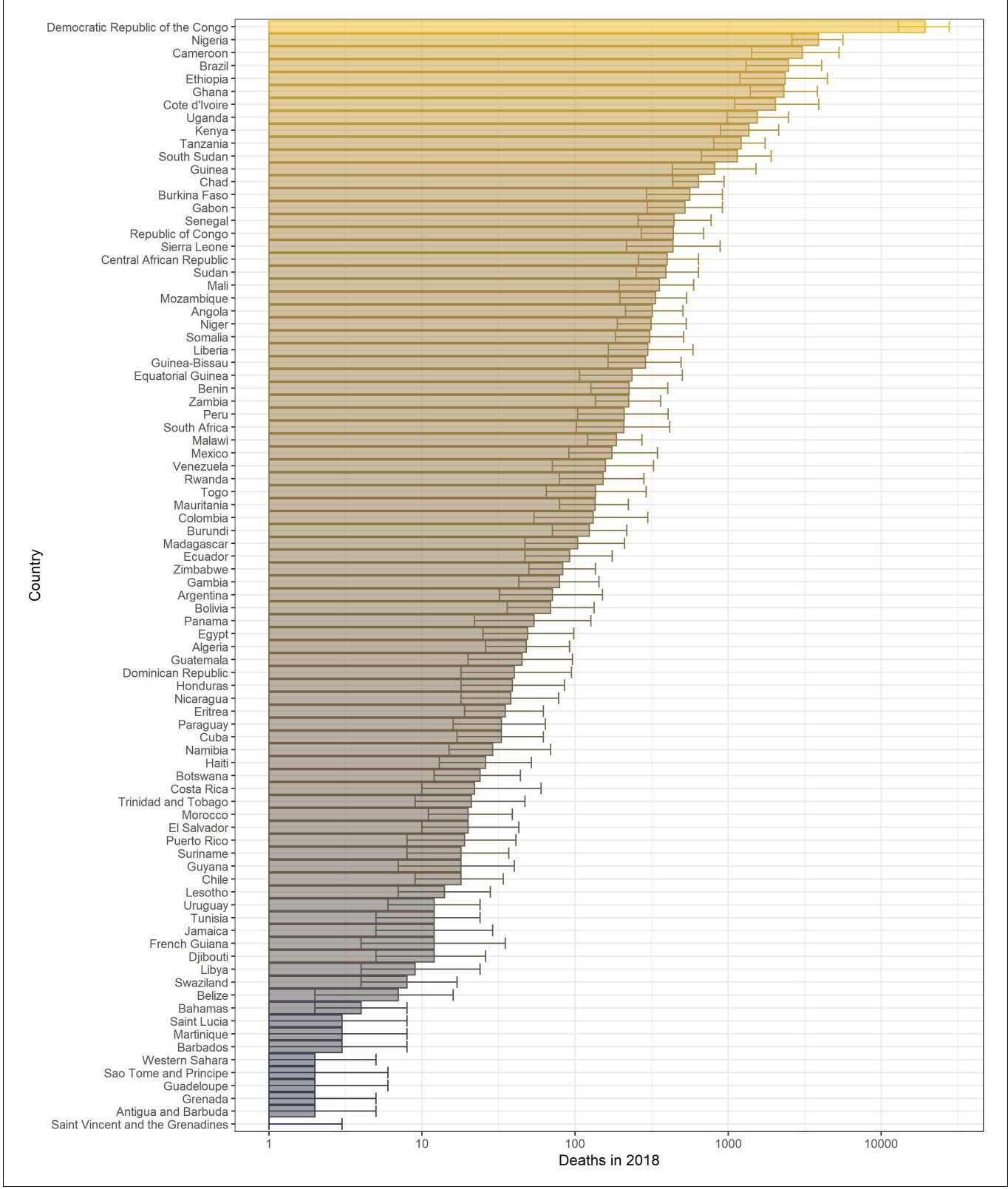

**Figure 8.** Posterior predicted potential deaths per country in 2018 from the ensemble model projections.

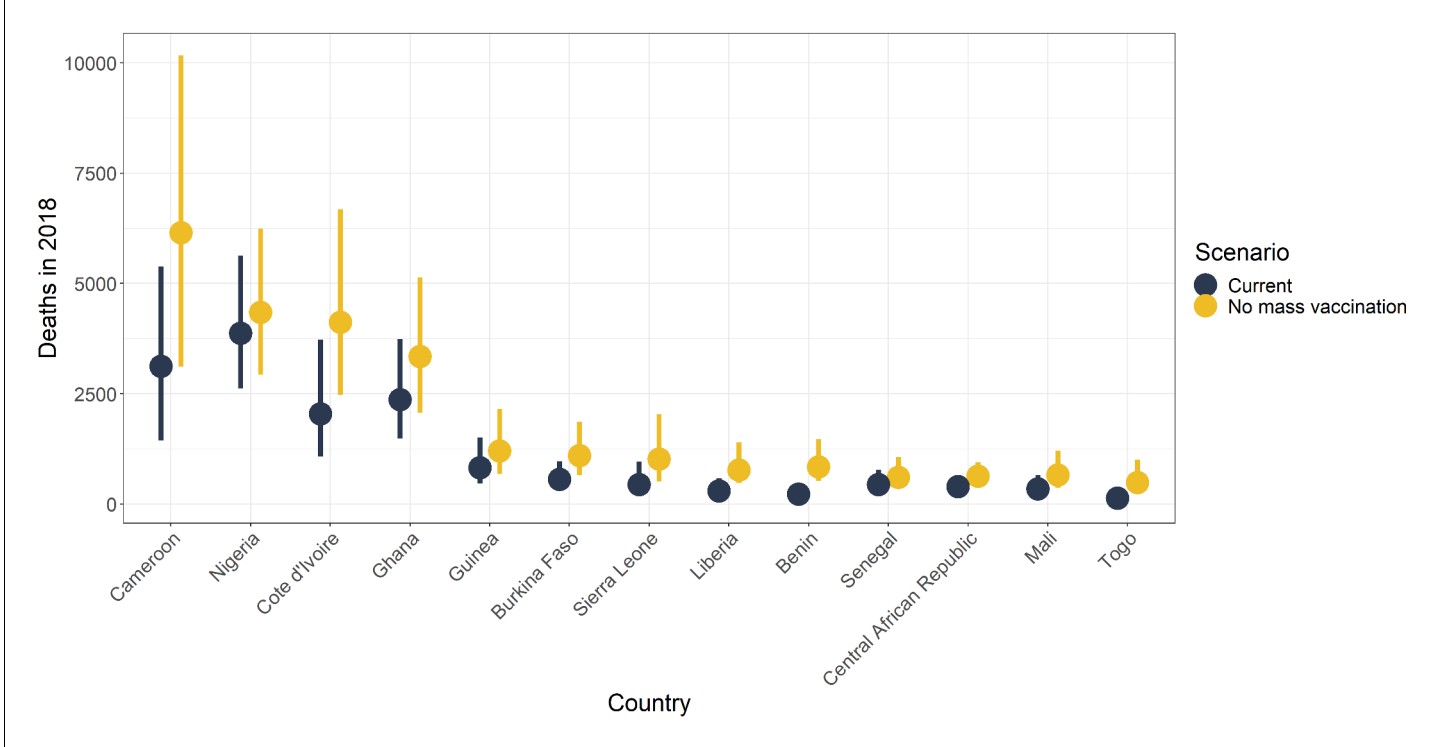

**Figure 9.** Median posterior predicted deaths averted for 2018 by country. Yellow represents the number of deaths without mass vaccination campaigns since 2006, and black represents deaths with current vaccination coverage levels. The points denote median and the line shows the 95% credible interval. See *Figure 9—figure supplement 1* for results in 2013.

The online version of this article includes the following figure supplement(s) for figure 9:

**Figure supplement 1.** Median posterior predicted deaths averted for 2013 by country.

estimates. In a similar way, vaccination activities are collated from a number of sources with all efforts made to ensure completeness. Yet there are activities that may have been omitted or not correctly parameterised could affect the results of this study. A dominant area of uncertainty is in the symptom spectrum of yellow fever. In our model, we estimate infections and then scale these to arrive at severe infections or cases and, finally, deaths. In this we use the estimates of Johannsson et al. to capture the uncertainty in the proportion of infections considered severe etc.; however, this remains an area of contention for yellow fever as previous estimates of case fatality ratio (CFR) have varied substantially and are significantly larger than other flaviviruses such as dengue (*Johansson et al., 2014*; *Oo et al., 2017*). The estimates of CFR we use also differ from those in the global burden of disease study leading to substantial differences in burden estimates between the two, although under-reporting is also addressed differently in the global burden of disease study (GBD) (*Compare, 2019*).

We have refined an established model but have also inherited some of its limitations, one of which is constancy. We assume that dynamics do not change substantially over the observation period in each province. As such, the variation over time is dominated by changes in demography and vaccination. In reality, the epidemiology of yellow fever is likely to change and has seen changes over recent years. Brazil experienced some of the largest outbreaks in its history with yellow fever in

**Table 4.** Deaths averted per year due to mass vaccination activites occurring from 2006 onwards in Africa.

| Year | Median deaths averted | Deaths averted, 95% CrI low | Deaths averted, 95% CrI high |
|------|----------------------|------------------------------|-------------------------------|
| 2013 | 11,414 | 6400 | 19,369 |
| 2018 | 10,140 | 5781 | 17,307 |

2017 and 2018; this was suggested to have been caused by changing patterns of human behaviour, such as urbanisation and movement, or changes in epidemiology in the sylvatic cycle; however, the full list of causes remains unclear (*Couto-Lima et al., 2017*; *Moreira-Soto et al., 2018*; *Chen et al., 2019*; *Possas et al., 2018*; *Saúde, 2019*). Spillover is also inherently stochastic, whereas, due to the focus on long-term burden, we assume a constant risk of spillover. As such, the model will not capture outbreak dynamics over a short time window but may highlight areas at most risk of outbreaks. The resulting estimates of burden and vaccine impact are thus the *potential* number of deaths given the conditions in each province and each country given the environmental conditions but may vary year on year due to outbreaks and stochastic spillover events.

## Conclusion

We have refined and extended an established model to update estimates of disease burden for yellow fever. We find consistent results that 92.2% (95% CrI [88.8–95%]) of global burden occurs in Africa and that mass vaccination activities have substantially reduced the number of cases and deaths we see today. We also highlight areas when burden is potentially high, in part due to lower-than-optimal vaccination coverages. The optimal route to avert deaths and potential yellow fever outbreaks is through tackling areas, and sub-populations, with low vaccination coverage. This is because vaccination is the main intervention for yellow fever, both as a preventative measure coupled with surveillance and as an outbreak response intervention. However, uncertainty in current data sources, and their interpretation, will limit the effectiveness of planning strategies. Our modelling approach underscores the need to examine background immunity, due to both natural infection and vaccination, in order to address not only the risk of future deaths but also assess how much of yellow fever is actually visible. For an old disease with an effective vaccine, yellow fever still poses new threats and, allowed to run unchecked, will provide a substantial health burden in many tropical areas as well as posing a significant global exportation risk.

## Additional information

### Funding

| Funder | Grant reference number | Author |
| --- | --- | --- |
| Bill and Melinda Gates Foundation | OPP1117543 | Katy AM Gaythorpe<br>Tini Garske<br>Neil Ferguson |
| Medical Research Council | MR/R015600/1 | Katy AM Gaythorpe<br>Arran Hamlet<br>Tini Garske<br>Neil Ferguson |
| Bill and Melinda Gates Foundation | | Katy AM Gaythorpe<br>Tini Garske<br>Neil Ferguson |

The funders had no role in study design, data collection and interpretation, or the decision to submit the work for publication.

### Author contributions

Katy AM Gaythorpe, Conceptualization, Data curation, Formal analysis, Validation, Investigation, Visualization, Methodology, Writing - original draft, Writing - review and editing; Arran Hamlet, Data curation, Formal analysis, Methodology, Writing - review and editing; Kévin Jean, Formal analysis, Methodology, Writing - review and editing; Daniel Garkauskas Ramos, Data curation, Validation, Methodology, Writing - review and editing; Laurence Cibrelus, Data curation, Writing - review and editing; Tini Garske, Conceptualization, Data curation, Supervision, Investigation, Methodology, Writing - review and editing; Neil Ferguson, Supervision, Methodology, Project administration, Writing - review and editing

**Author ORCIDs**
Katy AM Gaythorpe (ID) https://orcid.org/0000-0003-3734-9081
Kévin Jean (ID) https://orcid.org/0000-0001-6462-7185

**Decision letter and Author response**
Decision letter https://doi.org/10.7554/eLife.64670.sa1
Author response https://doi.org/10.7554/eLife.64670.sa2

# Additional files

### Supplementary files
• Transparent reporting form

### Data availability

Public repository data: Vaccination cov erage: coverage is available to download from the PoLiCi shiny app (https://shiny.dide.imperial.ac.uk/polici/). Serology surveys: There are seven published surveys used, available at DOI: 10.1016/0147-9571(90)90521-T , DOI: 10.1093/trstmh/tru086 , DOI: 10. 1186/s12889-018-5726-9 , DOI: 10.4269/ajtmh.2006.74.1078 , PMID: 3501739 , PMID: 4004378 , PMID: 3731366. Demographic data: Population level data was obtained from UN WPP (https://population.un.org/wpp/), this was disaggregated using Landscan 2017 data (https://landscan.ornl.gov/landscan-data-availability). Environmental data: This was obtained from LP DAAC (https://lpdaac.usgs.gov/) and worldclim (http://www.worldclim.org/). Non-human primate occurrence: This was obtained from the IUCN red list (https://www.iucnredlist.org/resources/spatial-data-download). Mosquito occurrence: This was obtained from The global compendium of Aedes aegypti and Ae. albopictus occurrence (https://doi.org/10.1038/sdata.2015.35). Yellow fever outbreaks: These were compiled from the WHO weekly epidemiologic record and disease outbreak news (https://www.who.int/wer/en/ and https://www.who.int/csr/don/en/). Compiled dataset for Africa available from https://github.com/kjean/YF_outbreak_PMVC/tree/main/formatted_data (copy archived at https://archive.softwareheritage.org/swh:1:rev:14703d7c5c7f63df6de04b81d5a48751604a906a/). Data elsewhere: The data from the WHO YF surveillance database and from recent serological surveys from WHO member states in Africa underlying the results presented in the study are available from World Health Organization (contact: William Perea, pereaw@who.int or Laurence Cibrelus, cibrelusl@who.int or Jennifer Horton, jhorton@who.int). Data from the occurrence of YF in Brazil were obtained from the Brazilian MoH (contact: Daniel Garkauskas Ramos). Code is available from: https://github.com/mrc-ide/YFestimation (copy archived at https://archive.softwareheritage.org/swh:1:rev:a352fe9369c4e1ec5915d63f0b36c8cfcc18894b/) for estimation and https://github.com/mrc-ide/YellowFeverModelEstimation2019 (copy archived at https://archive.softwareheritage.org/swh:1:rev:38f62f18a9261be7b987e042402331b6aa514233/) for specific analyses.

The following previously published dataset was used:

| Author(s) | Year | Dataset title | Dataset URL | Database and Identifier |
|---|---|---|---|---|
| Kraemer MUG, Sinka ME, Duda KA, Mylne A, Shearer FM, Brady OJ, Messina JP, Barker CM, Moore CG, Carvalho RG, Coelho GE, Van Bortel W, Hendrickx G, Schaffner F, Wint GRW, Elyazar IRF, Teng H-J, Hay SI | 2017 | The global compendium of Aedes aegypti and Ae. albopictus occurrence | https://datadryad.org/stash/dataset/doi:10.5061/dryad.47v3c | Dryad Digital Repository, 10.5061/dryad.47v3c |

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
