## [Decision Letter]

**Acceptance summary:**

This manuscript reports on a significant update to a leading model of the burden of yellow fever and the impact of vaccination. It provides estimates of the global burden of yellow fever in 2018 and the impact of vaccination activities in Africa. This paper is of interest to a broad range of researchers and public health practitioners engaged in the management of yellow fever.

**Decision letter after peer review:**

Thank you for submitting your article "The global burden of yellow fever" for consideration by *eLife*. Your article has been reviewed by three peer reviewers, including Jennifer Flegg as the Reviewing Editor and Reviewer #1, and the evaluation has been overseen by Miles Davenport as the Senior Editor. The following individual involved in review of your submission has agreed to reveal their identity: Alex Perkins (Reviewer #2).

Summary:

This manuscript reports on a significant update to a leading model of the burden of yellow fever and the impact of vaccination. Some of the major updates include extending it from Africa to also include South America, and accounting for model uncertainty with different combinations of spatial covariates. Some differences with this update of the model are reported and explained, and the impact of mass vaccination campaigns is quantified. This paper is of interest to a broad range of researchers and public health practitioners engaged in the management of yellow fever. It provides estimates of the global burden of yellow fever in 2018 and the impact of vaccination activities in Africa. Given the limited volume and quality of data available on yellow fever epidemiology, the modelling approach is appropriate and supports the study conclusions.

Essential Revisions:

1) Novelty.

The novelty of the method is oversold, particularly in the Abstract. It is not accurate to say that this manuscript "develops a novel framework" or "newly developed methodology" given that it is an update of an existing framework by Garske et al. The exact novelty of the framework should be made clear. Please cite the Garske et al. paper where you first mention the framework in the Materials and methods.

2) Data.

Insufficient details are provided on some model inputs. The authors should provide further details on the YF occurrence data used in the analysis. In particular, details on how cases were diagnosed, and discuss the potential impact of diagnostic uncertainties on the study results. Further details and justification should also be provided on the non-human primate covariate. For example:

– YF occurrence data. Please summarise the number of occurrence records used in the analysis by geographic region and time. Briefly summarise how occurrences of YF were diagnosed. PCR? Serology? Clinically? If serological and /or clinically diagnosed cases are included, please comment on the potential impact of misdiagnosed cases (e.g. due to cross-reactivity with other flaviviruses such as dengue) and how this may have impacted your results and conclusions.

– NHPs data. What is the difference between habitat suitability and occurrence of non-human primates? Also, I suggest that you do not refer to the species range maps provided by IUCN as "species distributions" as these are not modelled species distributions (as the authors point out in the Discussion). Perhaps refer to them as "range maps". Please clarify what you mean by "species richness". Did you count the number of species range maps covering at least 10% of each province? Which NHPs did you include and what was the justification?

– Vector data. Ae aegypti is the main vector for urban yellow fever transmission, please justify why you included Ae Albopictus as a potential covariate.

3) Materials and methods.

Some of the methods are unclear and it is not obvious which data is used to inform which model parameters. For example, it is unclear how were severe infections and deaths estimated from transmission intensity surface. Please provide details in the Materials and methods section. Consider including a workflow diagram to help the reader to follow more easily what you have done. Please justify the choice of priors (e.g. prior on force of infection seems fairly small and strong) as well as choice to sample proportional to AUC.

4) Presentation of results.

Please provide additional information to help readers understand the tables and figures without referring back to the text. For example:

– Table 1. Please add more informative column and row labels. Describe what 1s and 0s represent in the figure legend.

– Table 3. These are not all environmental covariates as the legend suggests – there appears to be reservoir and vector species included here. Please re-produce the table with more informative descriptions of each covariate and provide references.

– Table 4. Please define each parameter in the table legend.

– Figure 2. Please provide more descriptive labels for each plot.

– Figure 4. Are these estimates for 2018? Please state in the legend.

– Figure 5. Please provide more informative labels on each of the plots. Please describe what the dots and bars represent in the figure legend.

– Figure 6. Are these estimates for 2018? Please state in the legend.

The AUCs of the models are essentially all the same, in which case it doesn't look like there was much success in discriminating among them and the ensemble is essentially a simple average of the component models – can the authors comment?

Please define what you mean by "potential" deaths in the Materials and methods and/or Results section. You mention it in the Discussion, but it should be earlier. Conceptually this was a bit confusing because "potential" could be interpreted as assuming no vaccination.

5) Discussion – for the explanation for the discrepancy in force of infection spatially as compared to Garske et al. -- can we really be convinced that this is the correct interpretation (i.e., lower FOI in West Africa) or if we should consider this to be a sensitivity of the model?

6) Please discuss why you only estimated vaccination impact in Africa and not South America.

Reviewer #1:

This paper presents a mathematical estimation of the burden of yellow fever in Africa and South America, using multiple types of data. Results are presented based on ensemble model predictions. While the paper presents results of public health interest, I'm not convinced of the novelty of the approach, this is rather an extension of an existing methodology to more data and regions. For this reason, I feel like this work would be better suited in a specialist journal.

1) I'm not convinced that the contribution of this paper is novel enough for publication in *eLife*. The model framework was already largely in place, as was most of the data. I think the Abstract oversells the novelty of the methodology.

2) The first mention of a temporal component to the models was in the Results. I found the lack of introduction of the temporal nature of the models quite confusing.

3) What was done for serological surveys in South America, since there were none available? How were the serological data representative of the whole population? Can this be justified more?

4) It would be good to be clearer about which model parameters go where and which data is used to inform which parameters. E.g. a workflow diagram would help the reader to follow more easily what you have done.

5) Why is BIC used for model selection? That's not exactly a natural choice for Bayesian models since it does not consider the effect of the choice of priors.

Reviewer #2 :

Abstract – It is not accurate to say that this manuscript "develops a novel framework" or "newly developed methodology" given that it is an update of an existing framework by Garske et al. This issue is handled appropriately elsewhere in the manuscript, but here it is not.

Materials and methods – The prior on force of infection seems fairly small and strong. How was this choice made, and how sensitive are the results to it?

Materials and methods – For the ensemble predictions, is there a specific rationale or precedent for sampling proportional to AUC? It sounds reasonable, but also somewhat arbitrary without better justification.

Results – The AUCs of the models are essentially all the same, in which case it doesn't look like there was much success in discriminating among them and the ensemble is essentially a simple average of the component models. Am I missing I anything with that assessment?

Discussion – The explanation for the discrepancy in force of infection spatially as compared to Garske et al. is appreciated. I wonder if we can really be convinced that this is the correct interpretation (i.e., lower FOI in West Africa) or if we should consider this to be a sensitivity of the model. I'm not sure whether we can say without some sort of out of sample test of the predictions of these two models.

Reviewer #3:

Gaythorpe et al. estimated the global burden of yellow fever and the impact of mass vaccination activities in Africa in 2018. A previously published Bayesian modelling framework was extended and applied to a range of new and updated data sources. First, the authors updated an existing dataset of yellow fever occurrences (from 1987 to 2018). These data were used, along with a range of geospatial covariate data, to estimate the probability of yellow fever being reported in each first administrative region (i.e. province) within yellow fever risk zones. Measures of climatic and environmental variables and the presence of non-human primate reservoir species and mosquito vector species, among other factors, were included as model covariates. Data from a number of serological surveys was used to account for under-reporting and to estimate transmission intensity across the study region. Next, the authors updated an existing dataset of vaccination activities and used these data to calculate the number of deaths attributable to yellow fever at a province level, globally, and to estimate the number of deaths averted by vaccination in Africa. The authors estimated that in 2018, there were 51,000 (95%CrI[31,000-82,000]) deaths globally due to yellow fever, with 90% of the burden in Africa. Further, they estimated that vaccination averted 10,000 (95%CrI[6,000-17,000]) deaths in Africa in 2018. The study did not estimate the impact of vaccination in South America. The data available for studying global YF epidemiology is limited in volume and quality, which means that analyses such as the one presented here are inherently uncertain. This study considered uncertainty from estimation and model structure, but did not account for other key sources of uncertainty, i.e., in estimates of vaccination coverage or model covariates. Nonetheless, the study demonstrates a useful approach to estimating disease burden and vaccination impact over broad geographic areas. The data and approach seem appropriate to support the study's conclusions. However, in the manuscript's current state, some aspects of the analysis and model inputs need to be further clarified and justified:

1) The authors claim that they have developed a novel framework for estimating disease burden and vaccine impact. However, the study appears to make a number of extensions (e.g. incorporating new geographic regions, new covariates, updated data) to previously published methods. The authors should make the novelty of the framework more clear.

2) Limited details are provided on some model inputs. The authors should provide further details on the YF occurrence data used in the analysis. In particular, details on how cases were diagnosed, and discuss the potential impact of diagnostic uncertainties on the study results. Further details and justification should also be provided on the non-human primate covariate. For example, which NHP species were included and why.

3) The authors should provide information on the data and approach used to estimate the number of severe infections and deaths from the estimates of transmission intensity.

4) The authors claim that they developed a novel framework. However, the work appears to make a number of extensions (e.g. incorporating new geographic regions, new covariates, updated data) to previously published methods. Please make the novelty of the framework more clear.

5) Limited information on the model inputs are provided in the Materials and methods section. Where the authors point to previously published methods or dataset, they should at least provide a brief summary of the method/dataset. For updated or new datasets, more detailed descriptions should be provided. For example:

– YF occurrence data. Please summarise the number of occurrence records used in the analysis by geographic region and time. Briefly summarise how occurrences of YF were diagnosed. PCR? Serology? Clinically? If serological and /or clinically diagnosed cases are included, please comment on the potential impact of misdiagnosed cases (e.g. due to cross-reactivity with other flaviviruses such as dengue) and how this may have impacted your results and conclusions.

– NHPs data. What is the difference between habitat suitability and occurrence of non-human primates? Also, I suggest that you do not refer to the species range maps provided by IUCN as "species distributions" as these are not modelled species distributions (as the authors point out in the Discussion). Perhaps refer to them as "range maps". Please clarify what you mean by "species richness". Did you count the number of species range maps covering at least 10% of each province? Which NHPs did you include and what was the justification?

– Vector data. Ae aegypti is the main vector for urban yellow fever transmission, please justify why you included Ae Albopictus as a potential covariate.

6) How were severe infections and deaths estimated from transmission intensity surface? Please provide details in the Materials and methods section.

7) Please define what you mean by "potential" deaths in the Materials and methods and/or Results section. You mention it in the Discussion, but it should be earlier. Conceptually this was a bit confusing because "potential" could be interpreted as assuming no vaccination.

---

## [Author Response]

Essential Revisions:1) Novelty.The novelty of the method is oversold, particularly in the Abstract. It is not accurate to say that this manuscript "develops a novel framework" or "newly developed methodology" given that it is an update of an existing framework by Garske et al. The exact novelty of the framework should be made clear. Please cite the Garske et al. paper where you first mention the framework in the Materials and methods.

Many thanks for this, we have made sure to re-write the Abstract so that the methods are not oversold. We have also cited Garske et al. on first mention to make sure the framework is clearly explained.

In summary, this work applies the framework of Garske et al. to a new geographic region, South America, with updated datasets on yellow fever occurrence, substantially more serological survey studies and new environmental and species occurrence dataset. Within the Materials and methods we have made updates to accommodate the new structure and aims. We included the best fitting models in a hierarchical Bayesian framework and sampled proportionally from their posterior predictive distributions to accommodate both parameter and structural uncertainty in the resulting predictions.

2) Data.Insufficient details are provided on some model inputs. The authors should provide further details on the YF occurrence data used in the analysis. In particular, details on how cases were diagnosed, and discuss the potential impact of diagnostic uncertainties on the study results. Further details and justification should also be provided on the non-human primate covariate. For example:– YF occurrence data. Please summarise the number of occurrence records used in the analysis by geographic region and time. Briefly summarise how occurrences of YF were diagnosed. PCR? Serology? Clinically? If serological and /or clinically diagnosed cases are included, please comment on the potential impact of misdiagnosed cases (e.g. due to cross-reactivity with other flaviviruses such as dengue) and how this may have impacted your results and conclusions.

We have included further description on the occurrence data for YF including:

– linking to the outbreak dataset available on github for Africa: https://github.com/kjean/YF_outbreak_PMVC/tree/main/formatted_data

– Adding further detail on the diagnosis methods and the inclusion of suspected YF cases as a measure of surveillance effort.

– NHPs data. What is the difference between habitat suitability and occurrence of non-human primates? Also, I suggest that you do not refer to the species range maps provided by IUCN as "species distributions" as these are not modelled species distributions (as the authors point out in the Discussion). Perhaps refer to them as "range maps". Please clarify what you mean by "species richness". Did you count the number of species range maps covering at least 10% of each province? Which NHPs did you include and what was the justification?

Thank you, we have added further information detailing how the NHP data is included and correcting the language from “species distribution” to “range maps. The included NHP families are mentioned in Table 1 and all species within those families are included in the calculation of “species richness”, we have detailed this further in the data description.

– Vector data. Ae aegypti is the main vector for urban yellow fever transmission, please justify why you included Ae Albopictus as a potential covariate.

There are a number of vectors of YF, aegypti is the main urban vector. However, albopictus is also known to be able to carry YF. As such, we included both in the covariate selection process. We have noted this in the list of covariates. However, Ae. albopictus was not found to be significant in the final models.

3) Materials and methods.Some of the methods are unclear and it is not obvious which data is used to inform which model parameters. For example, it is unclear how were severe infections and deaths estimated from transmission intensity surface. Please provide details in the Materials and methods section. Consider including a workflow diagram to help the reader to follow more easily what you have done. Please justify the choice of priors (e.g. prior on force of infection seems fairly small and strong) as well as choice to sample proportional to AUC.

Thank you, we have made an effort to redress the Materials and methods section in this regard. Specifically:

– We have clarified that the estimates of number of severe infections and deaths come from Johannsson et al. and are used to scale the model output, infections.

– We have included a diagram to explain the data used at each stage (Figure 2)

– We have further explained the choice of priors. These were largely retained from earlier work such as Garske et al. and sensitivity analysis was performed therein. We have also included a further supplementary figure which shows the difference in force of infection estimates with two priors: the original exponential with rate = 0.001, and another exponential with rate = 0.1. The estimates extensively overlap- as such the results are not sensitive to this choice of prior. See new Figure 6—figure supplement 2.

– We have clarified sampling proportional to AUC although we found small differences if using AUC compared to, for example, likelihood.

4) Presentation of results.Please provide additional information to help readers understand the tables and figures without referring back to the text. For example:– Table 1. Please add more informative column and row labels. Describe what 1s and 0s represent in the figure legend.

Thank you, we have updated the table with better labels and further description in the legend.

– Table 3. These are not all environmental covariates as the legend suggests – there appears to be reservoir and vector species included here. Please re-produce the table with more informative descriptions of each covariate and provide references.

We changed the table to a bulleted list with a reference for each covariate data source.

– Table 4. Please define each parameter in the table legend.

We have updated the legend with parameter definitions.

– Figure 2. Please provide more descriptive labels for each plot.

We have updated the plot labels.

– Figure 4. Are these estimates for 2018? Please state in the legend.

We have added a description to the legend. The occurrence is a probability over the observation period 1984-2019.

– Figure 5. Please provide more informative labels on each of the plots. Please describe what the dots and bars represent in the figure legend.

Thank you, we have added further detail to the legend.

– Figure 6. Are these estimates for 2018? Please state in the legend.

We have added further detail to the legend – Force of infections are assumed to be time invariant as such, these do not correspond to a particular year.

The AUCs of the models are essentially all the same, in which case it doesn't look like there was much success in discriminating among them and the ensemble is essentially a simple average of the component models – can the authors comment?

We have added further description on the averaging. We found that although the AUC largely overlapped, there were some differences with where the bulk of the distribution lay. As such, and because these models were similar in performance, we chose to sample proportionately, thus including any bias, and providing a good reflection of the true structural uncertainty in the Results.

Please define what you mean by "potential" deaths in the Materials and methods and/or Results section. You mention it in the Discussion, but it should be earlier. Conceptually this was a bit confusing because "potential" could be interpreted as assuming no vaccination.

We have added a paragraph on this in the Materials and methods to further explain what we indicate by “potential”. We aimed to describe the fact that what we present captures an average burden over the years and does not, for example, include some of the stochastic effects that may lead to outbreaks or lack thereof.

5) Discussion – for the explanation for the discrepancy in force of infection spatially as compared to Garske et al. – can we really be convinced that this is the correct interpretation (i.e., lower FOI in West Africa) or if we should consider this to be a sensitivity of the model?

This may be a sensitivity of the model and we have added further to the Discussion to highlight this point. Particularly as data in West Africa (particularly serology data) is lacking which would be incredibly valuable in pinning down the exact magnitude of the force of infection in this region.

6) Please discuss why you only estimated vaccination impact in Africa and not South America.

We focused on the continuing effects of mass vaccination campaigns carried out in Africa, this is an updated analysis of Garske et al. to demonstrate the continued benefit of those campaigns.

Reviewer #1:This paper presents a mathematical estimation of the burden of yellow fever in Africa and South America, using multiple types of data. Results are presented based on ensemble model predictions. While the paper presents results of public health interest, I'm not convinced of the novelty of the approach, this is rather an extension of an existing methodology to more data and regions. For this reason, I feel like this work would be better suited in a specialist journal.1) I'm not convinced that the contribution of this paper is novel enough for publication in eLife. The model framework was already largely in place, as was most of the data. I think the Abstract oversells the novelty of the methodology.

We have rewritten the Abstract to avoid overselling the manuscript.

2) The first mention of a temporal component to the models was in the Results. I found the lack of introduction of the temporal nature of the models quite confusing.3) What was done for serological surveys in South America, since there were none available? How were the serological data representative of the whole population? Can this be justified more?

Unfortunately, there are no serological surveys available for South America partly due to concerns about cross-reactivity with other flaviviruses and partly as the vaccination coverage is generally high, rendering seroprevalence studies potentially less informative. We included serological data that includes information on the vaccination status of the included individuals and was not conducted as part of an outbreak investigation. As such, these provide a valuable view of the population of the province and time when the survey was conducted. We have added a further sentence on the serological survey inclusion.

4) It would be good to be clearer about which model parameters go where and which data is used to inform which parameters. E.g. a workflow diagram would help the reader to follow more easily what you have done.

Thank you, we have added a diagram (Figure 2) to illustrate which data is used where.

5) Why is BIC used for model selection? That's not exactly a natural choice for Bayesian models since it does not consider the effect of the choice of priors.

BIC is used in the covariate selection process only. AUC and likelihoods are examined to compare the final model variants.

Reviewer #2:Abstract – It is not accurate to say that this manuscript "develops a novel framework" or "newly developed methodology" given that it is an update of an existing framework by Garske et al. This issue is handled appropriately elsewhere in the manuscript, but here it is not.Materials and methods – The prior on force of infection seems fairly small and strong. How was this choice made, and how sensitive are the results to it?

See above for discussion and new figure (Figure 6—figure supplement 2) where we compare the estimates for two choices of prior on the force of infection.

Materials and methods – For the ensemble predictions, is there a specific rationale or precedent for sampling proportional to AUC? It sounds reasonable, but also somewhat arbitrary without better justification.

We examined both sampling by likelihood and by AUC, there was no major difference, likely due to the point below. As such, and because we were using AUC to compare against the original model of Garske et al., 2013, and the updated model of Gaythorpe et al., 2019, we chose to sample by AUC. This was also substantially more computationally efficient for the large sample sizes.

Results – The AUCs of the models are essentially all the same, in which case it doesn't look like there was much success in discriminating among them and the ensemble is essentially a simple average of the component models. Am I missing I anything with that assessment?

Whilst the distribution of AUC generally overlap, there are differences in where the majority of the distribution lies for example if we compare variant 1 and 2 we not that there are key differences. As such, and because we also did not want to omit potentially useful covariates, we chose to sample proportionally from all of the models.

Discussion – The explanation for the discrepancy in force of infection spatially as compared to Garske et al. is appreciated. I wonder if we can really be convinced that this is the correct interpretation (i.e., lower FOI in West Africa) or if we should consider this to be a sensitivity of the model. I'm not sure whether we can say without some sort of out of sample test of the predictions of these two models.

Thank you, this is a fair point and we have added to the Discussion to highlight that this may be a sensitivity of the approach, especially given that only occurrence data is available in West Africa rather than both occurrence and serology.

Reviewer #3:Gaythorpe et al. estimated the global burden of yellow fever and the impact of mass vaccination activities in Africa in 2018. A previously published Bayesian modelling framework was extended and applied to a range of new and updated data sources. First, the authors updated an existing dataset of yellow fever occurrences (from 1987 to 2018). These data were used, along with a range of geospatial covariate data, to estimate the probability of yellow fever being reported in each first administrative region (i.e. province) within yellow fever risk zones. Measures of climatic and environmental variables and the presence of non-human primate reservoir species and mosquito vector species, among other factors, were included as model covariates. Data from a number of serological surveys was used to account for under-reporting and to estimate transmission intensity across the study region. Next, the authors updated an existing dataset of vaccination activities and used these data to calculate the number of deaths attributable to yellow fever at a province level, globally, and to estimate the number of deaths averted by vaccination in Africa. The authors estimated that in 2018, there were 51,000 (95%CrI[31,000-82,000]) deaths globally due to yellow fever, with 90% of the burden in Africa. Further, they estimated that vaccination averted 10,000 (95%CrI[6,000-17,000]) deaths in Africa in 2018. The study did not estimate the impact of vaccination in South America. The data available for studying global YF epidemiology is limited in volume and quality, which means that analyses such as the one presented here are inherently uncertain. This study considered uncertainty from estimation and model structure, but did not account for other key sources of uncertainty, i.e., in estimates of vaccination coverage or model covariates. Nonetheless, the study demonstrates a useful approach to estimating disease burden and vaccination impact over broad geographic areas. The data and approach seem appropriate to support the study's conclusions. However, in the manuscript's current state, some aspects of the analysis and model inputs need to be further clarified and justified:1) The authors claim that they have developed a novel framework for estimating disease burden and vaccine impact. However, the study appears to make a number of extensions (e.g. incorporating new geographic regions, new covariates, updated data) to previously published methods. The authors should make the novelty of the framework more clear.

Thank you we have clarified this in the Abstract and have added caveats throughout the text.

2) Limited details are provided on some model inputs. The authors should provide further details on the YF occurrence data used in the analysis. In particular, details on how cases were diagnosed, and discuss the potential impact of diagnostic uncertainties on the study results. Further details and justification should also be provided on the non-human primate covariate. For example, which NHP species were included and why.

We have added further details on the occurrence data and the non-human primate covariates.

3) The authors should provide information on the data and approach used to estimate the number of severe infections and deaths from the estimates of transmission intensity.

We have included details on this and further referenced Johansson et al.

4) The authors claim that they developed a novel framework. However, the work appears to make a number of extensions (e.g. incorporating new geographic regions, new covariates, updated data) to previously published methods. Please make the novelty of the framework more clear.

Thank you, we have addressed this in both the Abstract and the Materials and methods sections.

5) Limited information on the model inputs are provided in the Materials and methods section. Where the authors point to previously published methods or dataset, they should at least provide a brief summary of the method/dataset. For updated or new datasets, more detailed descriptions should be provided. For example:– YF occurrence data. Please summarise the number of occurrence records used in the analysis by geographic region and time. Briefly summarise how occurrences of YF were diagnosed. PCR? Serology? Clinically? If serological and /or clinically diagnosed cases are included, please comment on the potential impact of misdiagnosed cases (e.g. due to cross-reactivity with other flaviviruses such as dengue) and how this may have impacted your results and conclusions.

We have included further description on the occurrence data for YF including:

– linking to the outbreak dataset available on github for Africa: https://github.com/kjean/YF_outbreak_PMVC/tree/main/formatted_data

– Adding further detail on the diagnosis methods and the inclusion of suspected YF cases as a measure of surveillance effort

– NHPs data. What is the difference between habitat suitability and occurrence of non-human primates? Also, I suggest that you do not refer to the species range maps provided by IUCN as "species distributions" as these are not modelled species distributions (as the authors point out in the Discussion). Perhaps refer to them as "range maps". Please clarify what you mean by "species richness". Did you count the number of species range maps covering at least 10% of each province? Which NHPs did you include and what was the justification?

Thank you, we have added further information detailing how the NHP data is included and correcting the language from “species distribution” to “range maps”. The included NHP families are mentioned in Table 1 and all species within those families are included in the calculation of “species richness”, we have detailed this further in the data description.

– Vector data. Ae aegypti is the main vector for urban yellow fever transmission, please justify why you included Ae Albopictus as a potential covariate.

There are a number of vectors of YF, Aegypti is the main urban vector. However, albopictus is also known to be able to carry YF. As such, we included both in the covariate selection process. We have noted this in the list of covariates. However, Ae. Albopictus was not found to be significant in the final models.

6) How were severe infections and deaths estimated from transmission intensity surface? Please provide details in the Materials and methods section.

Severe infections and deaths were calculated from the model output: infections scaled by the full uncertainty range of Johansson et al. who estimated the proportion of infections considered severe and, of those, the proportion of those who go on to die of YF. We have added further clarification in the text and in the flow diagram, Figure 2.

7) Please define what you mean by "potential" deaths in the Materials and methods and/or Results section. You mention it in the Discussion, but it should be earlier. Conceptually this was a bit confusing because "potential" could be interpreted as assuming no vaccination.

We have added a paragraph on this in the Materials and methods to further explain what we indicate by “potential”. We aimed to describe the fact that what we present captures an average burden over the years and does not, for example, include some of the stochastic effects that may lead to outbreaks or lack thereof.